# Evaluation of Plastic Deformation Considering the Phase-Mismatching Phenomenon of Nonlinear Lamb Wave Mixing

**DOI:** 10.3390/ma16052039

**Published:** 2023-03-01

**Authors:** Maoxun Sun, Yanxun Xiang, Wei Shen, Hongye Liu, Biao Xiao, Yue Zhang, Mingxi Deng

**Affiliations:** 1School of Mechanical Engineering, University of Shanghai for Science and Technology, Shanghai 200093, China; 2School of Mechanical and Power Engineering, East China University of Science and Technology, Shanghai 200237, China; 3School of Optical-Electrical and Computer Engineering, University of Shanghai for Science and Technology, Shanghai 200093, China; 4Shanghai Institute of Special Equipment Inspection and Technical Research, Shanghai 200062, China; 5School of Mechanical Engineering, Nantong University, Nantong 226019, China; 6College of Aerospace Engineering, Chongqing University, Chongqing 400044, China

**Keywords:** guided waves, nonlinear ultrasounds, wave mixing, phase mismatching, plasticity

## Abstract

Nonlinear guided elastic waves have attracted extensive attention owing to their high sensitivity to microstructural changes. However, based on the widely used second harmonics, third harmonics and static components, it is still difficult to locate the micro-defects. Perhaps the nonlinear mixing of guided waves can solve these problems since their modes, frequencies and propagation direction can be flexibly selected. Note that the phenomena of phase mismatching usually occur due to the lack of precise acoustic properties for the measured samples, and they may affect the energy transmission from the fundamental waves to second-order harmonics as well as reduce the sensitivity to micro-damage. Therefore, these phenomena are systematically investigated to more accurately assessing the microstructural changes. It is theoretically, numerically, and experimentally found that the cumulative effect of difference- or sum-frequency components will be broken by the phase mismatching, accompanied by the appearance of the beat effect. Meanwhile, their spatial periodicity is inversely proportional to the wavenumber difference between fundamental waves and difference- or sum-frequency components. The sensitivity to micro-damage is compared between two typical mode triplets that approximately and exactly meet the resonance conditions, and the better one is utilized for assessing the accumulated plastic deformations in the thin plates.

## 1. Introduction

Loads that exceed the yield stress limit cause irreversible plastic deformations in metals, and they can modify the mechanical behavior in chemical, nuclear, transportation, and aerospace industries [1,2]. Meanwhile, the microstructural evolution, such as the multiplication and motion of dislocations [3], deformation-induced phase changes, formation of the new phase [4], micro-crack initiation and growth [5], may occur during the plastic deformation. This undesirable inhomogeneous and localized damage on the microstructural level is usually regarded as the onset of the failure in the structures [6]. Therefore, quantitative evaluation of plasticity in metals, especially at their early stage, is important to ensure the structural safety and reliability. Nonlinear ultrasounds are sensitive to the microstructural defects whose sizes are much smaller than the wavelengths of the ultrasonic waves [7]. The generation of second harmonics has been widely applied to detect the plasticity in metals since they are relatively easy to generate in the experimental measurements [4,5,7,8,9,10]. However, especially for the dispersive and multimodal Lamb waves, the number of suitable mode pairs that exactly satisfy the resonance conditions are usually limited [11]. Recently, several novel methods were proposed to solve these problems and enhance the accuracy of the damage detection, e.g., second harmonics of S_0_ modes (lowest-order symmetric Lamb waves) at relatively low frequencies [12], the third harmonics [13] and the static components [14]. Similar trends, i.e., accumulated plasticity can lead to the monotonous increase of the acoustic nonlinearity parameters, were separately observed via the longitudinal waves [4,9], Rayleigh waves [15], and Lamb waves [8] since there is a fundamental relationship between the material inhomogeneity and plastic deformations, as discussed by Pruell et al. [8,16] and Shui et al. [9] However, the nonlinearities or signal noise interference in measurement systems also contribute to the generation of second harmonics, making it difficult to distinguish the source of the nonlinearities. Moreover, the spatial distribution of acoustic nonlinearity parameters over the region between the transmitter and receiver cannot be obtained via these methods [17,18].

For the nonlinear mixing of bulk or guided waves, the modes, frequencies, and propagation directions of the primary waves or generated harmonics can be flexibly selected based on the resonance conditions, so they can avoid the influence from nonlinearities of the measurement systems [19,20]. The relation of acoustic nonlinearity parameters and accumulated plasticity has been characterized by the nonlinear interaction of bulk waves, as observed by Croxford et al. [19] and Liu et al. [18] In addition, adjusting the position of the mixing zone can evaluate the material inhomogeneity in the area of interest. For example, the spatial distribution of the plasticity has been assessed by scanning the samples via the collinear [18,21] or non-collinear wave mixing [17], respectively. Further investigations of wave mixing are extended to the waveguides with the constant cross-sections, in which the dispersion and multimodal properties of the guided waves should be considered [20,22,23,24,25,26,27,28,29,30]. Moreover, these issues in composite laminates will become more complex due to the elastic anisotropy plus layered structure [31,32,33]. Deng [34] and de Lima et al. [22] theoretically confirmed that the cumulative second-order harmonics could be generated from the nonlinear interaction of continuous waves when the phase matching and non-zero power flux criteria are satisfied. Besides, when the fundamental waves and second-order harmonics travel in the different directions, vector analysis among them should be conducted as firstly pointed out by Hasanian et al. [23,24] and Ishii et al. [25] Moreover, Jiao et al. [26] and Metya et al. [27] used the nonlinear mixing of almost non-dispersive S_0_ modes for assessing the micro-cracks and localized deformation during creep in a steer plate. In addition, Lissenden’s group [20,23,24] systematically investigated the nonlinear mixing of horizontal shear waves via numerical simulations and experimental measurements, and they have characterized the localized fatigue [20] and thermal aging [23] in an aluminum plate. Meanwhile, for the guided waves in plates, experimental investigations of non-collinear and three-wave mixing [35,36,37] have attracted the interest of researchers in recent years. However, the resonance conditions may be not exactly satisfied due to the slight deviation from the phase matching in the practical applications or the absence of the precise material properties of the measured samples [38]. Therefore, it is necessary to investigate the phase-mismatching phenomenon in the nonlinear mixing of the guided waves. Especially, the second harmonics can be regarded as the sum-frequency components generated from the self-interaction of fundamental waves [24], corresponding studies of phase-velocity mismatching demonstrate that their amplitude remains bounded and oscillates with the spatial periodicity, and the spatial periodicity is a function of the fundamental frequencies, phase velocities of the fundamental waves and second harmonics, respectively [11,22,34,38]. Furthermore, S_0_ mode at a relatively low frequency that approximately meets the phase and group velocity matching conditions has been used to nondestructively characterize plasticity [12], micro-cracks [39], and imperfect joints [40]. However, for the more general case, i.e., the phase-mismatching phenomena in the nonlinear mixing of the guided waves, few reports are available for the theoretical, numerical, and experimental investigations. The aim of our work is to systematically investigate this phenomenon via theoretical analysis, finite element (FE) simulations and experiments, and develop a simple method that can avoid the influence from the external nonlinearity and accurately evaluate the early damages in the thin plate.

The remainder of this articles is organized as follows. In Section 2, the phenomenon of phase mismatching is theoretically investigated based on nonlinear mixing of the guided waves. Next, FE simulations and experiments are conducted to study the beat or cumulative effect of components at difference or sum frequencies in Section 3 and Section 4. It is demonstrated that cumulative components at difference and sum frequencies may be more suitable for assessing the microstructural changes. In Section 5, the accumulated plasticity is characterized by a mode triplet that exactly satisfies the phase matching conditions. The results are discussed in Section 6. Finally, conclusions are summarized in Section 7.

## 2. Theoretical Background

### 2.1. Phenomena of Phase Mismatching

For the nonlinear mixing of continuous waves propagating in the same directions, according to the perturbation approach and normal-mode expansion technique, the wave field of difference- and sum-frequency components can be expressed as a linear combination of the propagating modes at the frequencies $\omega_{a}\pm \omega_{b}$ [22]
(1)$$u^{\pm }(x,y,t)=\frac{1}{2}\sum_{m=1}^{\infty }A_{m}(x)U_{m}^{\pm }(y)exp[-j(\omega_{a}\pm \omega_{b})t],$$
where *x* is the propagation direction of the primary waves, *y* is the direction perpendicular to the surface, j is the imaginary unit. In addition, the subscripts a and b indicate the primary waves, *m* denotes the mode of difference or sum-frequency components. For the *m*-th mode at the frequencies $\omega_{a}\pm \omega_{b}$, $U_{m}^{\pm }(y)$ denotes the through-thickness displacements profiles. The amplitudes $A_{m}(x)$ of *m*-th mode at difference or sum frequency satisfy the following equations [22]
(2)$$4P_{mn}(\frac{d}{dx}-jk_{n}^{*})A_{m}(x)=g_{n}exp[j(k_{a}\pm k_{b})x],$$
(3)$$P_{mn}=-\frac{1}{4}\int_{-h}^{h}\left(\frac{v_{n}^{*}}{2}\cdot \frac{\sigma_{m}}{2}+\frac{v_{m}}{2}\cdot \frac{\sigma_{n}^{*}}{2}\right)\cdot n_{x}dy,$$
(4)$$g_{n}=\frac{1}{2}\int_{-h}^{h}v_{n}^{*}\cdot [\nabla \cdot \sigma^{\left(2\right)}(\nabla u_{a},\nabla u_{b},2)]\;dy-\frac{1}{2}v_{n}^{*}\cdot \sigma^{\left(2\right)}(\nabla u_{a},\nabla u_{b},2)\cdot n_{y}|{}_{-h}^{h}.$$
where $k_{n}^{*}$, $k_{a}$ and $k_{b}$ are the wavenumbers of second-order harmonics, primary wave a and b, $n_{x}$ and $n_{y}$ are the unit vector in the *x* and *y* direction, the first Piola-Kirchhoff stress **σ** consists of the linear parts **σ**^(1)^ and nonlinear parts **σ**^(2)^. For $k_{n}^{*}-(k_{a}\pm k_{b})\neq 0$, the solution of the amplitude $A_{m}(x)$ can be written as
(5)$$A_{m}(x)=\frac{g_{n}}{2P_{mn}\Delta k}\sin(\frac{\Delta kx}{2})\exp[j(k_{n}^{*}-\frac{\Delta k}{2})x],$$
where $\Delta k=k_{n}^{*}-(k_{a}\pm k_{b})$. Based on the Equation (5), if $P_{mn}\neq 0$ and $g_{n}\neq 0$, the amplitude $A_{m}(x)$ remains bounded and oscillates with the spatial periodicity $L_{p}$, which equals to
(6)$$L_{p}=|\frac{2\pi }{\Delta k}|.$$

For the special case $k_{a}=k_{b}$, the spatial periodicity $L_{p}$ of the sum-frequency components (i.e., the so-called second harmonics) is given by $L_{p}=|\frac{2\pi }{k_{2a}-2k_{a}}|$, as reported in the references [11,22,34]. Meanwhile, the difference-frequency components can be regarded as the so-called static components. When $k_{n}^{*}\to k_{a}\pm k_{b}$, since $\underset{\Delta k\to 0}{\lim}[2\sin(\Delta kx/2)/\Delta k]=x$, the solution of the amplitude $A_{m}(x)$ can be written as
(7)$$A_{m}(x)=\frac{g_{n}}{4P_{mn}}x\exp[j(k_{a}\pm k_{b})x].$$

The amplitude $A_{m}(x)$ of difference- or sum-frequency components increases linearly with the propagation distance when $P_{mn}\neq 0$ and $g_{n}\neq 0$. Note that $P_{mn}\neq 0$ is regarded as a necessary criterion, to ensure that the generated components at difference or sum frequency are the propagating modes. In addition, the condition $g_{n}\neq 0$ guarantees non-zero power flux from the primary waves to difference- or sum-frequency components. In summary, if all resonance conditions are satisfied, the nonlinear mixing of symmetric (or antisymmetric) modes can only lead to the generation of the symmetric mode. Moreover, the antisymmetric mode can only be generated from the nonlinear interaction of symmetric and antisymmetric modes.

Nonlinear mixing of the Lamb waves will be utilized to characterize the changes of material inhomogeneities. Hence, for the FE simulations and experimental measurements, it is necessary to build a parameter which only depends on the material parameters of the measured samples. According to the Equation (7), the displacements of the difference- or sum-frequency components on the surface can be expressed as $Re\{(xg_{n}/8P_{mn})U_{m}^{\pm }(\pm \frac{h}{2})$$\exp[j(k_{a}\pm k_{b})x-j(\omega_{a}\pm \omega_{b})t]\}$. Meanwhile, the product of the displacements of primary wave a and b on the surface can be written as $Re\{\frac{1}{2}U_{a}(\pm \frac{h}{2})$$\exp[j(k_{a}x-\omega_{a}t)]\}$Re{12Ub(±h2)exp[j(kbx−ωbt)]}. Therefore, the corresponding displacement ratio recorded on the surface at the fixed position (i.e., $y=\pm \frac{h}{2}$ and $x=x_{0}$) equals to
(8)$$\frac{Amp[u_{m}^{\pm }(x_{0},\pm \frac{h}{2},t)]}{Amp[u_{a}(x_{0},\pm \frac{h}{2},t)]Amp[u_{b}(x_{0},\pm \frac{h}{2},t)]}=\frac{x_{0}g_{n}}{2P_{mn}}\frac{U_{m}^{\pm }(\pm \frac{h}{2})}{U_{a}(\pm \frac{h}{2})U_{b}(\pm \frac{h}{2})},$$
where $Amp[u_{m}^{\pm }(x_{0},\pm \frac{h}{2},t)]$, $Amp[u_{a}(x_{0},\pm \frac{h}{2},t)]$ and $Amp[u_{b}(x_{0},\pm \frac{h}{2},t)]$ are the amplitudes of displacements at difference or sum frequency and fundamental frequencies, respectively. For the selected mode triplets with fixed frequencies, the displacement ratio in Equation (8) ratio is only a function of the material parameters, including the density, Young’s modulus, Poisson’s ratio and third-order elastic constants (TOECs), etc. Therefore, the acoustic nonlinearity parameter can be established based on the Equation (8) [20,24,26,28,30,36]
(9)$$\chi =\frac{A_{\pm }}{A_{a}A_{b}},$$
in which $A_{\pm }$, $A_{a}$ and $A_{b}$ are amplitudes of the difference- or sum-frequency components and fundamental waves in the frequency domain. It is noted that the changes of micro-defects (e.g., dislocations, micro-cracks, or micro-voids) may lead to variations of the Young’s modulus [41] and TOECs. In addition, the acoustic nonlinearity parameter *χ* shows high sensitivity to the changes of TOECs. Thus, the proposed *χ* probably has the potential to evaluate the micro-damages in the samples.

### 2.2. Selection of Mode Triplets

Lager mixing zones are probably required for investigating the phase-mismatching phenomena in nonlinear mixing of Lamb waves since the energy transfer from the primary waves to the difference- or sum-frequency components only occur in the mixing zone. Therefore, the energy can be continuously transferred when the tone bursts a and b travel in the same directions and possess the similar group velocities. S_0_ modes at relatively low frequencies attract much attention since they have approximately equal phase and group velocities. Single S_0_ modes can be excited due to their almost non-dispersive properties, and they can be utilized to explore the phenomena of phase mismatching. S_0_ modes at 400 kHz and 600 kHz is chosen as the fundamental waves in the first mode triplet, as shown in Figure 1a. In addition, mode triplets that exactly meet the phase matching criterion can be selected from the Lamb waves at the longitudinal wave speed, since their frequencies equal to integer multiples of $f_{L}=\frac{c_{T}}{h\sqrt{1-(c_{T}/c_{L}{)}^{2}}}$, where h is the thickness of the thin plate, $c_{T}$ and $c_{L}$ are the velocities of the transverse and longitudinal elastic waves [42]. S_2_ and S_3_ modes are chosen as the primary waves in the second mode triplet so that their difference and sum frequencies are different to those of higher harmonics, as illustrated in Figure 1b. Meanwhile, S_2_ and S_3_ modes have the largest group velocities at the frequencies $f=2f_{L}$ and $f=3f_{L}$, and perhaps they can be easily separated from the other modes.

## 3. Numerical Investigation of Nonlinear Lamb Waves Mixing

The proposed mode triplets that approximately and exactly satisfy the resonance conditions are firstly verified in numerical simulations. For nonlinear mixing of S_0_ modes at 400 kHz and 600 kHz, two-dimensional models are built using the commercial finite-element analysis software Abaqus 6.13/EXPLICIT, as shown in Figure 2. The thickness of the plate is set as 1 mm. The material of the plate is chosen as the 6061-T6 aluminum alloy, whose mass density, young’s modulus, and Passion’s ratio are 2704 kg/m^3^, 7.05 × 10^10^ Pa and 0.36, respectively. Corresponding third-order elastic constants (TOECs) defined by the Landau and Lifshitz [43] (their relations with TOCEs used by Murnaghan [44] can be written as l=B+C, $m=\frac{1}{2}A+B$ and n=C) are also involved here, in which A, B, and C equal to −4.16 × 10^11^ Pa, −1.31 × 10^11^ Pa, and −1.51 × 10^11^ Pa, respectively [45,46,47]. The plate is discretized by the four-node plane strain (CPE4R) elements with the maximum size of 0.02 mm. The infinite elements (CINPE4) are also applied on the right boundary so that the wave mixing would not be affected by the boundary reflections. The simulation time step is 1 × 10^−9^ s. Note that U1 and U2 denote the displacement components in *x* and *y* direction respectively.

For the nonlinear mixing of S_0_ modes at 400 kHz and 600 kHz, the fundamental waves are simultaneously excited by applying the uniform longitudinal displacements on the left end of the plate. The excitation signal is a linear superposition of the 10-cycle and 15-cycle Hanning windowed sinusoidal tone bursts at 400 kHz and 600 kHz, and their frequency spectra are given in Figure 3a. Besides, the primary wave a or b is individually excited by imposing the uniform longitudinal displacements on the left end of the plate with the 10-cycle or 15-cycle Hanning windowed sinusoidal tone bursts at 400 kHz or 600 kHz, respectively. The displacement component U2 at 140 mm are recorded when both primary a and b are excited (denoted as Case A), and transferred into the frequency domain using the Fast Fourier Transform (FFT), as shown in Figure 4a,b Besides fundamental waves, components at difference, double and sum frequency also appears in Figure 4b. Note that difference- and sum-frequency components are probably generated from the nonlinear mixing of guided waves. To confirm this assumption, typical time-domain signals and corresponding frequency spectra are also obtained at the same position when primary a and b are separately excited (denoted as Case B and Case C). It is founded that only the second harmonics exist in propagating Lamb waves, as shown in Figure 5a,b. Therefore, the source of the difference- and sum-frequency components can be clearly identified. Next, the modes of the generated harmonics are confirmed based on the group velocities and wave structures (i.e., the normalized distribution of the displacement component U1 and U2 along *y* axis), which are compared with the theoretical values obtained by the software DISPERSE v2.0.20f [48]. The displacement component U1 and U2 on the surface are received from 80 mm to 200 mm at intervals of 20 mm. To obtain the difference- and sum-frequency components, the signals are then processed by the band-pass filters with a central frequency of 200 kHz and 1 MHz. These filtered signals are then utilized to calculate the group velocity. Relevant values of components at difference and sum frequency are 5450.66 m/s and 4951.27 m/s. Compared with the theoretical group velocities of S_0_ mode at 200 kHz (5465.54 m/s) and 1 MHz (5128.8 m/s), their errors are calculated as 0.27% and 3.46% respectively. In addition, the displacement component U1 and U2 are obtained along the thickness at intervals of 0.1 mm. These signals are band-pass filtered to obtain the difference- and sum-frequency components. Amplitudes of these components are then extracted from the filtered signals and normalized with the maximum. The distributions of their amplitude along the thickness agrees well with the wave structures of S_0_ mode at 200 kHz and 1 MHz respectively.

For the second mode triplet that exactly satisfies the resonance conditions, the fundamental waves are simultaneously excited by prescribing the linear superposition of displacements $\frac{U_{a}(y)\sin(2\pi f_{a}t)[1-\cos(\frac{2\pi f_{a}t}{20})]}{2}$ and $\frac{U_{b}(y)\sin(2\pi f_{b}t)[1-\cos(\frac{2\pi f_{b}t}{30})]}{2}$ at the left end of the plate, where $U_{a}(y)$ and $U_{b}(y)$ are the displacement profiles of S_2_ mode at 6.99 MHz and S_3_ mode at 10.49 MHz [49]. The excitation signals are also analyzed by the FFT, as given in Figure 3b. Figure 6a, b show the typical displacement components U1 at *x* = 140 mm and corresponding frequency spectra of the windowed wave packages. According to the different group velocities, S_2_ and S_3_ modes are well separated from other Lamb modes which are effectively suppressed. However, only the components at fundamental frequencies can be clearly identified in the frequency domain. The sum-frequency components do not appear, which may be overwhelmed by the unwanted noise since the amplitude of them are relatively small. Hence, a pulse-inversion technique [50] is utilized to accentuate the difference- and sum-frequency components and remove the primary waves and second harmonics. Four ultrasonic measurements should be conducted in this signal-processing method. For the Case I and Case II, the excited signals of primary wave a and b have the same phase, and they equal to 0° and 180° respectively. In addition, the phase of excited signal is chosen as 0° for primary wave a and 180° for primary wave b in Case III, while the phases of primary waves in Case IV are exactly opposite. The ultrasonic signals measured in Case I, Case II, Case III, and Case IV are denoted as Signal I, Signal II, Signal III, and Signal IV respectively, then these signals will be processed based on Equation (10) [50]
(Signal I + Signal II − Signal III − Signal IV)/4.(10)
Note that only the difference- and sum-frequency components can exist after the signal processing, as illustrated in Figure 6c,d.

To calculate the group velocity of difference- and sum-frequency components, the displacement component U1 and U2 on the surface are received at intervals of 20 mm. Next, they are processed by the pulse-inversion technique and band-pass filters with a central frequency of 3.5 MHz or 17.48 MHz. The group velocities of difference- and sum-components are calculated as 3895.8 m/s and 3905.8 m/s, which almost equal to the theoretical value (3934.2 m/s). Meanwhile, the displacement component U1 and U2 are recorded along the thickness in steps of 0.04 mm. To obtain the corresponding amplitudes of difference and sum-frequency components, these original signals are then processed by the pulse-inversion technique and band-pass filters. The distribution of amplitude of displacement components U1 and U2 along the thickness are also compared with the wave structure obtained by the software DISPERSE [48], and it is found that these results coincide with the counterparts of S_1_ mode at 3.5 MHz and S_5_ mode at 17.48 MHz.

The sensitivity of two mode triplets to micro- or early damage are investigated by changing the value of TOECs. Several multiples (a = 0.5, 1, 2, 4) of the initial value are utilized in numerical simulations. The acoustic nonlinearity parameters $\frac{A_{-}}{A_{a}A_{b}}$ and $\frac{A_{+}}{A_{a}A_{b}}$ of displacement component U1 at 140 mm are then normalized by the counterparts using a = 0.5. Finally, the sensitivity to micro- or early damage is assessed by comparing the slopes of corresponding normalized acoustic nonlinearity parameters.

## 4. Experimental Investigation of Nonlinear Lamb Waves Mixing

The phenomena of phase mismatching are also studied based on the experimental observation. The high-power gate amplifier (RAM-5000 SNAP), high power attenuator, wedge transducers, pre-amplifier, oscilloscope, and computer are applied to generate and record the ultrasonic signals. Relevant wedge transducers consist of the longitudinal wave transducer and Plexiglas wedge.

Firstly, the mode triplet that approximately meets the resonance conditions, i.e., the nonlinear mixing of S_0_ mode at 400 kHz and 600 kHz can generate the S_0_ mode at 1 MHz, will be investigated in experiments. The linear superposition of the 10-cycle tone burst at 400 kHz and 15-cycle tone burst at 600 kHz is fed into the wedge transducer with the oblique angle of 30° and central frequency of 500 kHz, to simultaneously excite the primary wave a and b. The FFT is applied on the excitation signals to obtain the frequency spectra, which are shown in Figure 7a. The wedge transducer with an oblique angle of 30° and central frequency of 250 kHz (or 1 MHz) is utilized as the receiver, which is more sensitive to the component at difference frequency (or sum frequency). These wedge transducers are coupled to the surface of the 1-mm-thick 6061-T6 aluminum alloy plate. It is noted that a special fixture is designed to ensure the collinear arrangement of transmitters and receivers, as illustrated in Figure 8. Moreover, the contact pressure between the wedge transducers and sample remains constant, making it possible to conduct repeatable measurements and obtain reliable data. Finally, these original signals will be processed by the pulse-inversion technique and FFT. The typical processed signal received by the wedge transducer with the central frequency of 250 kHz at 150 mm and its frequency spectrum are shown in Figure 9a,b, respectively.

Secondly, experimental investigations are also conducted using the mode triplets that exactly meet the resonance conditions, i.e., the S_1_ mode at 3.5 MHz and S_5_ mode at 17.48 MHz can be generated from the nonlinear interaction of S_2_ mode at 6.99 MHz and S_3_ mode at 10.49 MHz. The excited signals consist of 120-cycle sinusoidal tone bursts at 6.99 MHz and 180-cycle sinusoidal tone bursts at 10.49 MHz respectively, and they are fed into the wedge transducer with an angle of 24.5° and central frequency of 10 MHz. The excitation signals are also analyzed by the FFT, which are given in Figure 7b. Meanwhile, the wedge transducers with the same angle are utilized to receive the components at the difference and sum frequency, while their central frequencies as selected as 3.5 MHz and 10 MHz respectively. The angle of wedge transducer is calculated as 24.5° based on Snell’s law and they are only sensitive to the modes whose phase velocities roughly equal to that of the longitudinal wave. However, for the Lamb modes whose phase velocity is similar to that of a longitudinal wave, perhaps they cannot be removed since there is a specific “bandwidth” in the “band-pass filter” and thus the unwanted modes with the similar phase velocities will also be excited. Fortunately, these unwanted modes can be separated with primary waves according to the different group velocities when the propagation distance is long enough. In addition, the components at difference and sum frequency generated by the unwanted modes perhaps can be neglected since the resonance conditions are not satisfied. Finally, the pulse-inversion technique and FFT will be used to extract the amplitudes of fundamental waves, difference- and sum-frequency components in received signals. The typical time-domain signals received by the wedge transducer with the central frequency of 10 MHz at 150 mm are processed and shown in Figure 10a,b.

## 5. Characterization of the Plastic Deformation in a Thin Plate

To study the influence of the phase-mismatching phenomena on the damage detection, specimens with different plastic deformation will be assessed by the mode triplet that is more sensitive to the microstructural changes. Six dog-bone samples were cut from one single 1-mm-thick plate along the rolling direction, and they are made of 6061-T6 aluminum alloy. According to national standard of China (GB/T 228.1-2010), the width and length of the gauge section are selected as 35 and 61.71 mm respectively. Acoustic nonlinearity parameters of these undamaged samples were obtained before the plastic deformation, and relevant measurements of each sample are repeated 5 times to calculate the average values. To produce various levels of plastic strains in gauge sections, the rest of the samples were subjected to different tensile loads above the yield stress limit via the servo-hydraulic testing machine (Instron 8803). The stress-strain curve of 6061-T6 aluminum alloy is shown in Figure 11, in which the maximum engineering strains of different samples (0%, 1.3%, 2.6%, 3.9%, 6.5% and 7.8%) are marked. Next, Acoustic nonlinearity parameters of these damaged samples were measured again, and they were then normalized to the values of pristine samples without damage.

## 6. Results and Discussion

### 6.1. Effect of Phase Mismatching in Numerical Simulations

According to the theoretical analysis, the acoustic nonlinearity parameters of components at difference and sum frequency may oscillate with the propagation distance when the condition of synchronism is not exactly satisfied. These phenomena will also be numerically investigated in this section. To establish the relationship of the acoustic nonlinearity parameters and propagation distance, the time-domain signals recorded on the surface from 80 mm to 200 mm at intervals of 20 mm are utilized again. They are processed by the FFT to extract the amplitudes of fundamental waves *A*_a_ or *A*_b_, the amplitudes of difference-frequency components *A*_−_, the amplitudes of sum-frequency components *A*_+_ and the amplitudes of second harmonics *A*_2a_ or *A*_2b_. The acoustic nonlinearity parameters can be written as $\frac{A_{2a}}{A_{a}^{2}}$, $\frac{A_{2b}}{A_{b}^{2}}$, $\frac{A_{-}}{A_{a}A_{b}}$, and $\frac{A_{+}}{A_{a}A_{b}}$ respectively. The acoustic nonlinearity parameters $\frac{A_{-}}{A_{a}A_{b}}$ and $\frac{A_{+}}{A_{a}A_{b}}$ of U1 and U2 versus the propagation distance are presented in Figure 12, in which the points and their fitting curves are obtained from the numerical data. The acoustic nonlinearity parameter $\frac{A_{+}}{A_{a}A_{b}}$ of U1 and U2 remains bounded and oscillates with the same spatial periodicity. The relevant values equal to 350 mm, which are close to the theoretical results calculated by Equation (6), i.e., L_p_ = 357.28 mm. However, the acoustic nonlinearity parameter $\frac{A_{-}}{A_{a}A_{b}}$ of U1 and U2 approximately exhibit a monotonic increasing trend with the propagation distance because their much longer spatial periodicities are around 2021.95 mm. In addition, there is a beat effect for the acoustic nonlinearity parameter $\frac{A_{2a}}{A_{a}^{2}}$ and $\frac{A_{2b}}{A_{b}^{2}}$ with the propagation distance, as shown in Figure 13a,b. The acoustic nonlinearity parameters $\frac{A_{2a}}{A_{a}^{2}}$ or $\frac{A_{2b}}{A_{b}^{2}}$ in Case 1 (i.e., primary waves a and b are simultaneously excited) are compared with the corresponding values in Case 2 or Case 3 (i.e., primary wave a and b are separately excited). Relevant spatial periodicity almost equals to each other and agrees well with the expected value obtained by Equation (6).

### 6.2. Effect of Phase Mismatching in Experimental Measurements

The experimental investigations of phase-mismatching phenomena are also carried out, in which the distance of the transmitter and receiver is varied from 100 mm to 500 mm in steps of 100 mm. For the nonlinear mixing of S_0_ mode at 400 kHz and 600 kHz, the time-domain signals received on the surface of plates are processed via the pulse-inversion technique and FFT. The extracted amplitudes of the components at difference frequencies, fundamental frequencies, and sum frequency, i.e., *A*_−_, *A*_a_, *A*_b_ and *A*_+_, are applied for establishing the acoustic nonlinearity parameters. Figure 14 presents the relationship between the propagation distances and acoustic nonlinearity parameters that are normalized by the $\frac{A_{+}}{A_{a}A_{b}}$ at 100 mm. It is noticed that $\frac{A_{+}}{A_{a}A_{b}}$ exhibits a beat effect with propagation distance while $\frac{A_{-}}{A_{a}A_{b}}$ grows monotonously with propagation distance, and they are then fitted to Equation (6). Moreover, the corresponding spatial periodicity of the fitting curves is consistent with the numerical results and theoretical prediction. For the mode triplets that cumulative S1 mode at 3.5 MHz and S5 mode at 17.48 MHz can be generated from the nonlinear mixing of S2 mode at 6.99 MHz and S3 mode at 10.49 MHz, the steps are adjusted to 50 mm. The pulse-inversion technique and FFT are also applied on the time-domain signals, to extract the amplitudes of components at difference, fundamental and sum frequencies, i.e., *A*_−_, *A*_a_, *A*_b_ and *A*_+_. The built $\frac{A_{-}}{A_{a}A_{b}}$ and $\frac{A_{+}}{A_{a}A_{b}}$ are then normalized to the counterparts at 150 mm. As shown in Figure 15a,b, both the normalized $\frac{A_{-}}{A_{a}A_{b}}$ and $\frac{A_{+}}{A_{a}A_{b}}$ increase linearly with propagation distance.

### 6.3. Characterization of Plasticity in the Thin Plates

As shown in Figure 16, for the same variations of TOECs, the slope of acoustic nonlinearity parameters of second mode triplet is larger than 3.5, which is approximately twice that of the first mode triplet. It is probably demonstrated that the cumulative components show high sensitivity to the microstructural changes in plates [51]. Thus, the mode triplet that satisfies the resonance conditions, i.e., the nonlinear mixing of S_2_ mode and S_3_ mode can generate the S_1_ mode, are used to assess the plastic deformation.

Both the pristine samples and damaged samples with different plastic deformation are measured by the selected mode triplet. The relationship of the normalized acoustic nonlinearity parameters and strain are shown in Figure 11. A dramatic increase of normalized acoustic nonlinearity parameters (around 50%) is observed at an early stage of plasticity (between 0% and 3.9%), followed by a gradual increase before the tensile strain reaches the value of 7.8%. It is reported that a number of dislocations can be generated during the plastic deformation of aluminum alloy [52], meanwhile significantly deformed grain, micro-void, and micro-cracks may also contribute to the variations of the normalized acoustic nonlinearity parameters [5].

## 7. Conclusions

Theoretical, numerical, and experimental investigations are conducted to systematically explore the phase-mismatching phenomena of nonlinear Lamb waves mixing. At first, an acoustic nonlinearity parameter $\frac{A_{\pm }}{A_{a}A_{b}}$ is established, which is only determined on the material parameters, including the density, Young’s modulus, Poisson’s ratio, and third-order constants. In addition, a signal processing method, i.e., pulse-inversion technique, is introduced to extract the amplitude of fundamental waves and difference- or sum-frequency components. Meanwhile, two typical mode triplets that approximately and exactly meet the phase matching criterion are selected for numerically and experimentally studying the phase-mismatching phenomena, i.e., nonlinear mixing of S_0_ modes at 400 kHz and 600 kHz, as well as nonlinear interaction of S_2_ mode and S_3_ mode with the same phase velocities equal to those of longitudinal waves.

It is theoretically found that the acoustic nonlinearity parameters increase linearly with propagation distance when resonance conditions are exactly satisfied, while they oscillate with the specific spatial periodicity $L_{p}=|\frac{2\pi }{\Delta k}|$ that only depends on the deviation from the phase matching criterion. The beat effects of difference-frequency, double-frequency, and sum-frequency components are also numerically and experimentally observed, meanwhile their spatial period is in good agreement with their theoretical counterparts. In addition, there is a cumulative effect for the difference- and sum-frequency components in the nonlinear mixing of S_2_ and S_3_ modes, which is more sensitive to the same microstructural changes by comparing with the first mode triplet. Therefore, difference-frequency components in the second mode triplet are utilized to assess the accumulated plasticity. It is indicated that the acoustic nonlinearity parameters $\frac{A_{-}}{A_{a}A_{b}}$ increase dramatically during 0 < strain ≤ 3.9% and grow slightly when 3.9 < strain ≤ 7.8%, and this trend should be attributed to the variations of dislocation density in the aluminum alloy plates.

## Figures and Tables

**Figure 1 materials-16-02039-f001:**
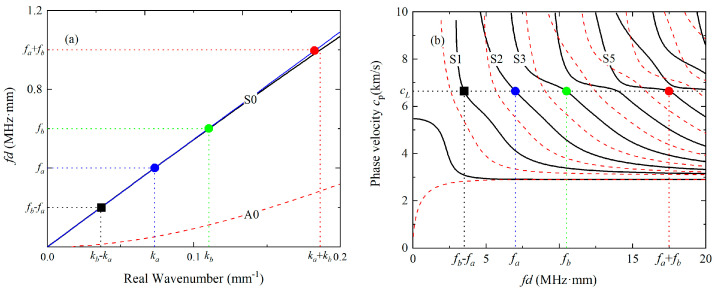
Dispersion curves for the 6061-T6 aluminum alloy plate. The first (**a**) and second (**b**) mode triplet that approximately and exactly satisfies the phase matching criterion are marked.

**Figure 2 materials-16-02039-f002:**
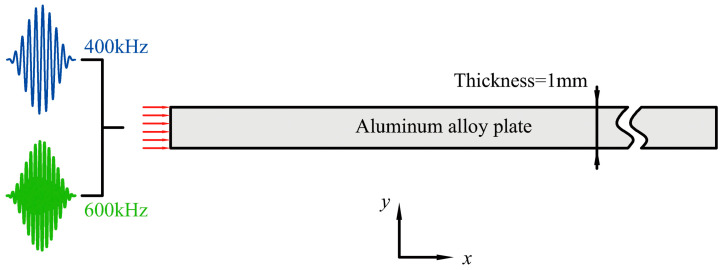
Schematic diagram of two-dimensional finite-element model for nonlinear mixing of S_0_ modes at 400 kHz and 600 kHz.

**Figure 3 materials-16-02039-f003:**
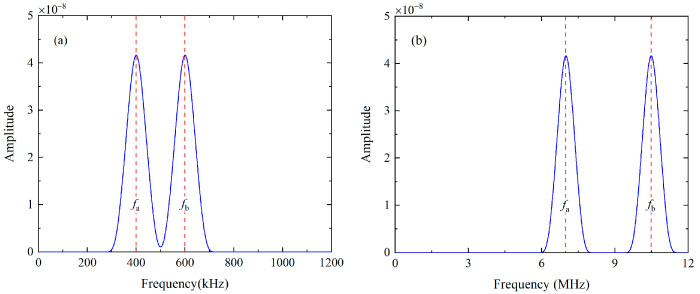
Frequency spectra of the excitation signals in the first (**a**) and second (**b**) mode triplet.

**Figure 4 materials-16-02039-f004:**
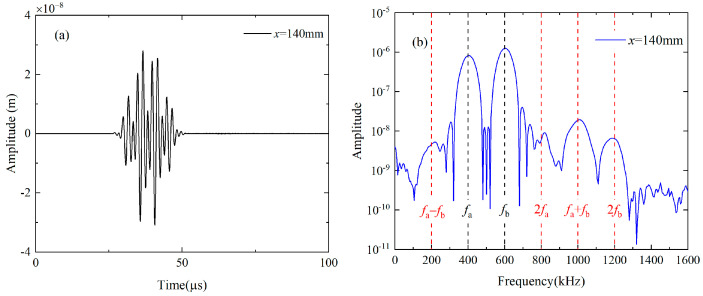
Typical displacement along *y* axis (**a**) received on the surface of the plate at *x* = 140 mm in Case 1 and its frequency spectrum (**b**) obtained by the FFT.

**Figure 5 materials-16-02039-f005:**
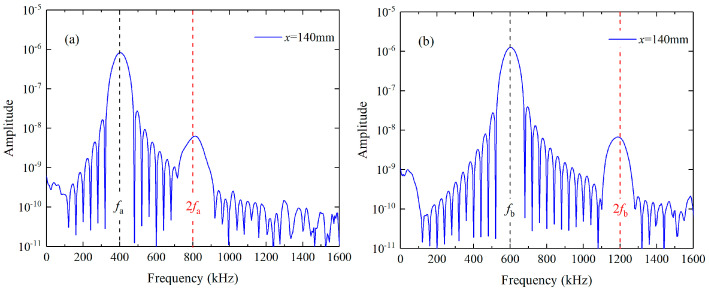
Frequency spectra of displacements along *y* axis at the same position in Case 2 (**a**) and Case 3 (**b**).

**Figure 6 materials-16-02039-f006:**
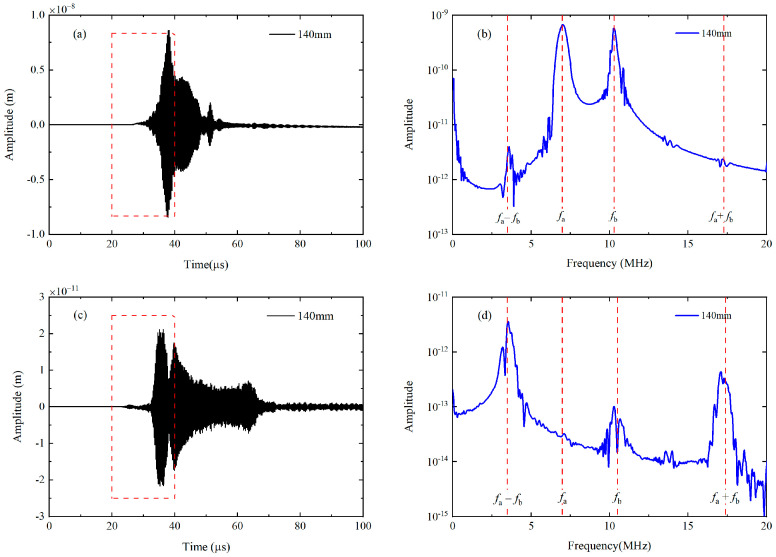
As for the second mode triplet in simulations, typical time-domain waveforms at *x* = 140 mm (**a**) within the time window (during 20 μs and 40 μs) are transferred into frequency domain (**b**) via the FFT; In addition, a group of time-domain signals within the time window (from 20 μs to 40 μs) are processed by the pulse-inversion technique (**c**) and the FFT (**d**).

**Figure 7 materials-16-02039-f007:**
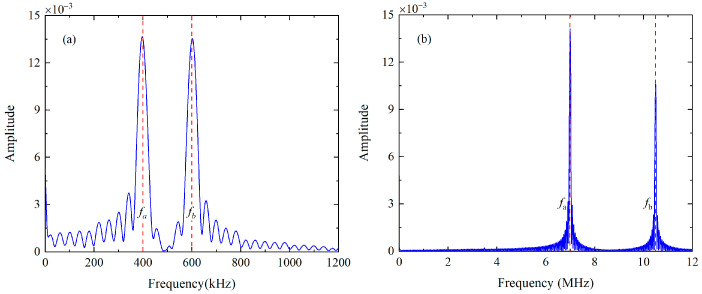
The excitation signals of the first (**a**) and second (**b**) mode triplet in experiments are also processed by the FFT.

**Figure 8 materials-16-02039-f008:**
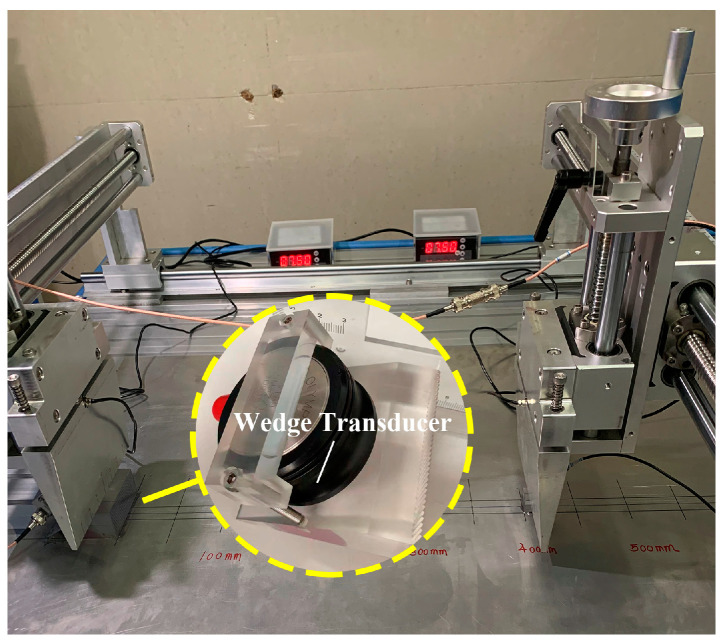
A designed fixture to ensure the collinear arrangement of transmitters and receivers, and maintain the constant contact pressure between two wedge transducers and plates.

**Figure 9 materials-16-02039-f009:**
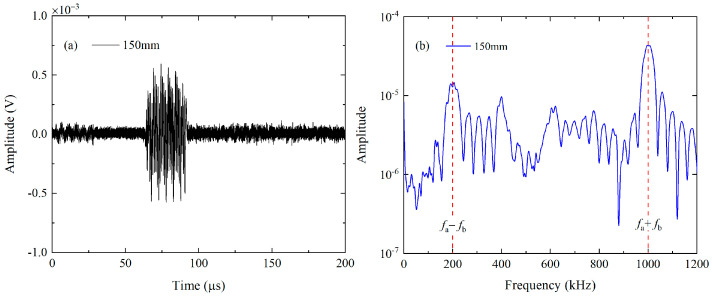
For the first mode triplet, four time-domain signals at 150 mm are processed by the pulse-inversion technique (**a**) and FFT (**b**).

**Figure 10 materials-16-02039-f010:**
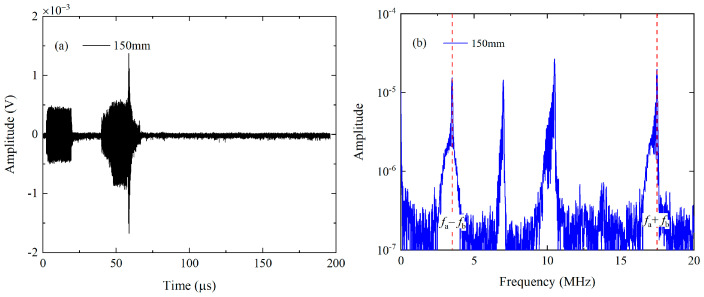
The processed time-domain signal (**a**) of the second mode triplet at 150 mm and its frequency spectra (**b**).

**Figure 11 materials-16-02039-f011:**
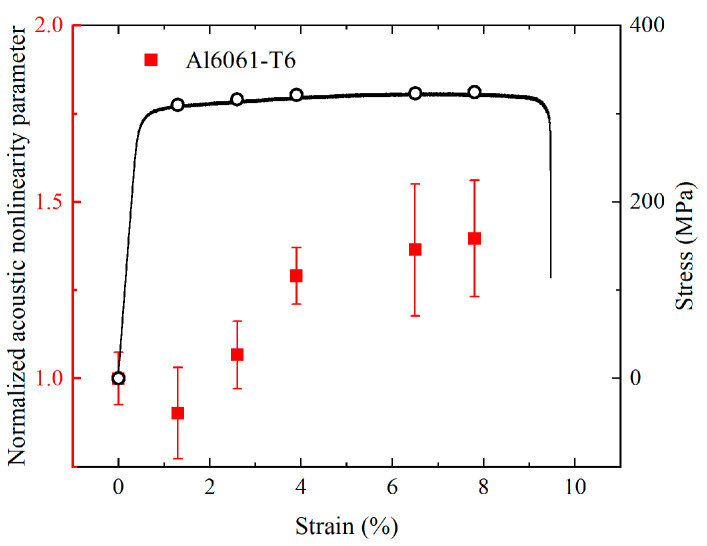
Relationship of acoustic nonlinearity parameters $\frac{A_{-}}{A_{a}A_{b}}$ and longitudinal engineering strain, superimposed on stress-strain curve of 6061-T6 aluminum alloy with maximum engineering strains for six samples marked.

**Figure 12 materials-16-02039-f012:**
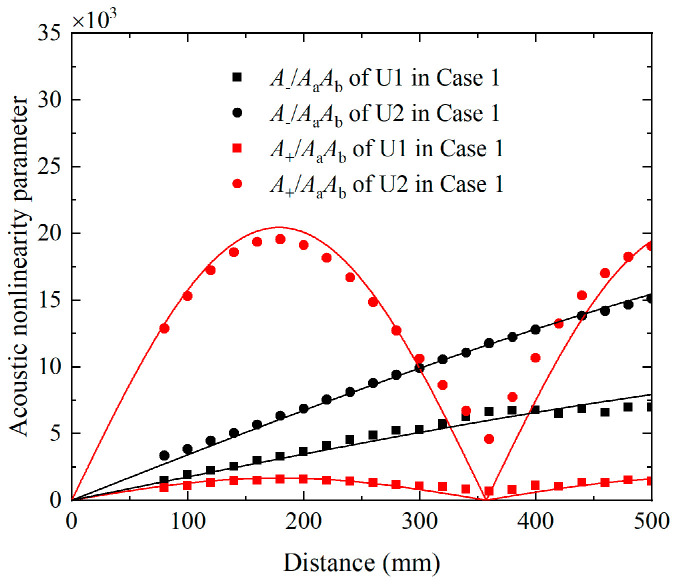
Acoustic nonlinearity parameters of difference-frequency components $\frac{A_{-}}{A_{a}A_{b}}$ and sum-frequency components $\frac{A_{+}}{A_{a}A_{b}}$ in *x* and *y* direction versus the propagation distance for Case 1, which are fitted by Equation (6).

**Figure 13 materials-16-02039-f013:**
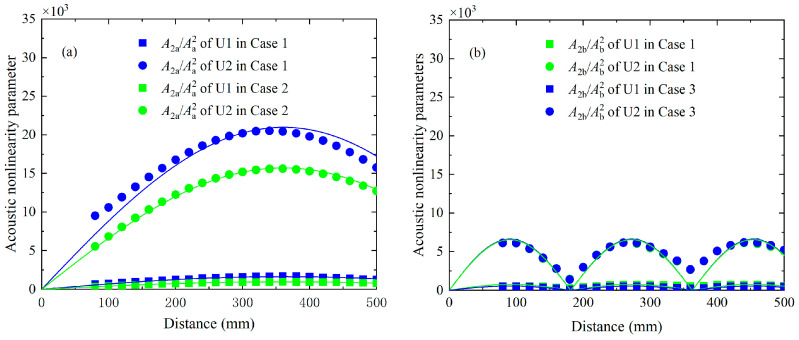
Acoustic nonlinearity parameters of second harmonics $\frac{A_{2a}}{A_{a}^{2}}$ in Case 1 and Case 2 (**a**), as well as $\frac{A_{2b}}{A_{b}^{2}}$ in Case 1 and Case 3 (**b**) as a function of propagation distance, which are also fitted by Equation (6).

**Figure 14 materials-16-02039-f014:**
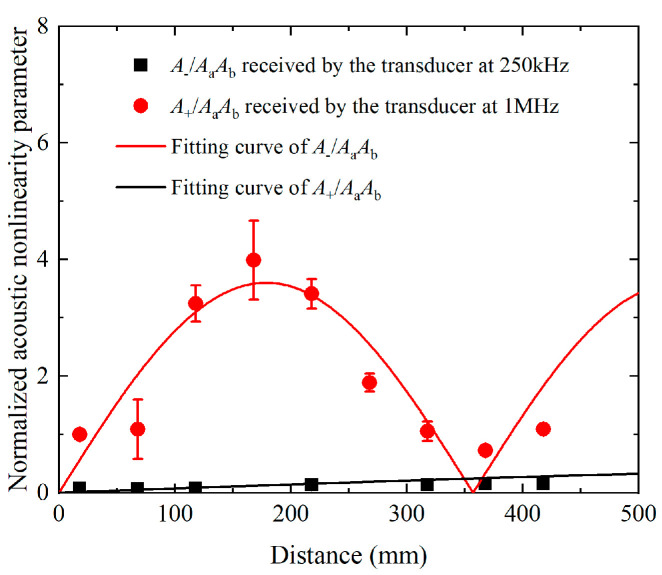
For the nonlinear mixing of S_0_ modes at 400 kHz and 600 kHz, normalized acoustic nonlinearity parameters of $\frac{A_{-}}{A_{a}A_{b}}$ or $\frac{A_{+}}{A_{a}A_{b}}$ and their fitting curve versus propagation distance, measured by the receivers with central frequencies of 250 kHz and 1 MHz respectively.

**Figure 15 materials-16-02039-f015:**
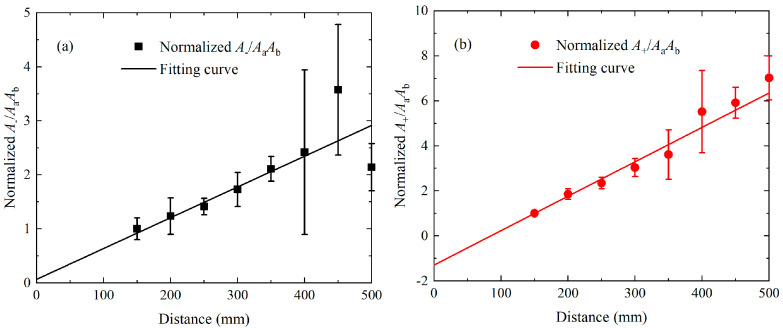
For the nonlinear interaction of S_2_ and S_3_ mode in experiments, the relationship of normalized acoustic nonlinearity parameters $\frac{A_{-}}{A_{a}A_{b}}$ (**a**) or $\frac{A_{+}}{A_{a}A_{b}}$ (**b**) and propagation distance, as well as their fitting lines.

**Figure 16 materials-16-02039-f016:**
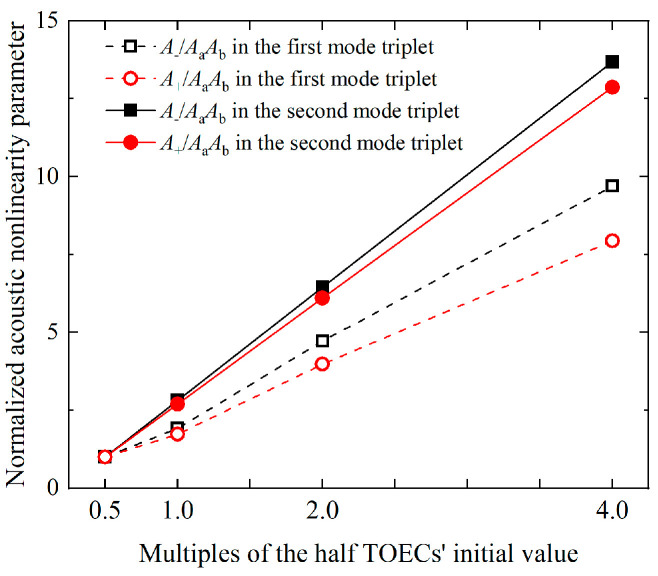
Variation of normalized acoustic nonlinearity parameters versus multiples of the half TOECs’ initial value.

## Data Availability

Not applicable.

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
