# Peer review of "Evaluation of Plastic Deformation Considering the Phase-Mismatching Phenomenon of Nonlinear Lamb Wave Mixing"

_materials, 2023, doi:10.3390/ma16052039_

Round 1

Reviewer 1 Report

The assessment of plasticity in a thin plate taking into consideration the phase-mismatching phenomena that occurs during nonlinear Lamb wave mixing is presented in this work. The findings of the study presented in the article are rather intriguing; hence, following a few simple revisions, the work need to be taken into consideration for publishing.

1. The application of this study should be briefly added in Abtract.

2. The author's introduction is pretty well written. The related works are reviewed quite comprehensively. However, at the end of this section, the authors should describe the structure of the paper so that it is easy for the reader to follow. In addition, some qualitative crack studies can be added to increase the abundance of this section as follows.

"Finite element modeling for free vibration response of cracked stiffened FGM plates"

"Finite element modeling of the bending and vibration behavior of three-layer composite plates with a crack in the core layer"

3. Some subscripts (such as: a, b, m, n) in equation (1) need to be defined.

4. Equation (8) needs to be cited.

5. Dimensions in the model in figure 2 should be shown for easy visualization.

6. The results and discussion sections are well presented.

In addition, the authors should review the manuscript carefully to avoid some typographical and grammatical errors.

Author Response

Response Letter

(Manuscript ID: materials-2221448)

    Thanks very much for the valuable comments on the manuscript materials-2221448 with the title of " Evaluation of plasticity in a thin plate considering the phase-mismatching phenomenon of nonlinear Lamb wave mixing". According to referee’s comments, we made some revisions and supplements. The responses to the referees are listed as followings.

(1) Comments:The application of this study should be briefly added in Abtract.

Reply: Thank you very much for the comment. As suggested, the application of this manuscript is briefly introduced.

More details can see Line 21-23 in Page 1.

(2) Comments:The author's introduction is pretty well written. The related works are reviewed quite comprehensively. However, at the end of this section, the authors should describe the structure of the paper so that it is easy for the reader to follow. In addition, some qualitative crack studies can be added to increase the abundance of this section as follows.

"Finite element modeling for free vibration response of cracked stiffened FGM plates"

"Finite element modeling of the bending and vibration behavior of three-layer composite plates with a crack in the core layer"

Reply: Thanks for the comment. As suggested, the organization of this paper is described at the end of Section 1. Meanwhile, two papers are added in the revised manuscript.

More details can see Line 103-111 in Page 3.

(3) Comments:Some subscripts (such as: a, b, m, n) in equation (1) need to be defined.

Reply: Thanks, it’s a good comment. As suggested, the definitions of these subscripts in Equation (1) are added.

(4) Comments:Equation (8) needs to be cited.

Reply: Thanks. Equation (8) are independently derived by the authors of this manuscript.

(5) Comments:Dimensions in the model in figure 2 should be shown for easy visualization.

Reply: Thanks. As suggested, to better display the finite-element model of nonlinear mixing of S0 modes, the dimensions in Figure 2 have been revised and added.

(6) Comments:The results and discussion sections are well presented.

Reply: Thanks. Figure 16 is changed in the revised manuscript since the first mode triplet and second mode triplet were confused in initial figure legend.

Finally, we have carefully checked our manuscript and made some revisions.

Thanks once again.

Reviewer 2 Report

The authors presented an experimental, theoretical and numerical studies of material micro-damage detection in Al6061-T6 aluminium alloy thin plate (1 mm thickness), using nonlinear ultrasound guided elastic waves of frequencies 400 kHz and 600 kHz, which satisfy the elastic resonance conditions, to generate a mixed 1000 kHz elastic signal. The evolution of material micro-damage, by performing plastic deformations in tensile tests of simple tensile specimens deformed in the rolling direction, were evaluated by the response of elastic waves applied to the microstructure of the specimen.

However, the authors state that it was difficult to locate the micro-defects in the thin plate, particularly the important major micro-defect which can grow and cause material failure. Moreover, the size and characteristics of the defect were also not investigated. The authors monitored and measured the signal amplitude A of the acoustic sound and defined the acoustic nonlinearity as the signal amplitude ratio A±/(Aa.Ab) and correlated with the plastic strain level. The authors reported the correlation between the specimen engineering plastic strain and the normalized acoustic nonlinearity ratio and they stated that “…found that the acoustic nonlinearity parameters increase dramatically during 0 < strain ≤ 3.9% and grow slightly when 3.9 < strain ≤ 7.8%.”.

This present work is based mainly in the article by Pruell et al. (2009): A nonlinear-guided wave technique for evaluating plasticity-driven material damage in a metal plate. NDT&E International 42 (2009) 199–203.

Furthermore, the literature review in the introduction is missing the work by Lemaitre (1985), modeling the development of material damage due to plastic deformations:

Lemaitre, J., 1985, A continuous damage mechanics model for ductile fracture. Journal of Engineering Materials and Technology. vol. 107: 83-89.

Therefore, the presented method to evaluate the development of material micro-damage due to plastic deformations by ultrasound waves are somewhat limited. For future work, I would suggest to correlate the signal amplitude ratio A±/(Aa.Ab), which satisfy the elastic resonance conditions, with the experimental variation of Young’s elastic modulus due to plastic deformations.

Please correct for better paper quality and clarity:

1.   Title, I suggest to correct “Evaluation of plasticity damage in a thin plate, using the phase-mismatching phenomenon of nonlinear Lamb elastic wave mixing”.

2.   Abstract, line 15, please correct “Nonlinear guided elastic waves…”. Also, line 29, please correct “…assessing the accumulated plastic deformations in the thin plates...”.

3.   Introduction, pg.1, please correct “Loads that exceed the yield stress limit cause irreversible plastic deformations…”. Also, line 37, “…may occur during the plastic deformation.”. Also, line 38, “…These undesirable inhomogeneous and localized damage...”.

4.   Introduction, pg.2, line 47, please correct “Recently, several novel methods were proposed…”. Also, correct “were” throughout the text. Also, line 48, “…So mode (lowest-order symmetric Lamb elastic waves)…”. Also, correct “Lamb elastic waves” throughout the text.

5. Introduction, pg.2, line 53, please correct “…between the material inhomogeneity and plastic deformations…”. Please correct “material inhomogeneity” throughout the text. Also, line 54, please correct “However, the nonlinearities or signal noise interference in measurement systems…”. Also, “…as discussed by Pruell et al. [8,9] and…” please include the reference:

[9] Pruell et al. (2009): A nonlinear-guided wave technique for evaluating plasticity-driven material damage in a metal plate. NDT&E International 42 (2009) 199–203.

6.  Theoretical background, pg.3, line 115, please correct “…denotes the through-thickness displacements…”. Also, pg.5, line 155, “…may lead to the variations of the Young’s modulus [1] and TOECs.”. Also, pg.5, line 166, “…excited due to their almost…”.

  [1] Lemaitre, J., 1985, A continuous damage mechanics model for ductile fracture. Journal of Engineering Materials and Technology. vol. 107: 83-89.

7.  Theoretical background, pg.5, line 172, please correct “…longitudinal elastic waves [37].”

8.  Numerical investigation of nonlinear Lamb waves mixing, pg.5, line 192, “Corresponding third-order elastic constants (TOECs) are also involved here, in which A, B, and C are equal to…” please define these material elastic constants ?

9.   Characterization of the plastic deformation in a thin plate, pg.11, line 346, please correct Six dog-bone samples were cut from…”. Also, line 350, “…of these undamaged samples were obtained…”. Also, line 353, “…samples were subjected to...”. Also, “...above the yield stress limit via…”. Also, line 355, “…maximum engineering strains of…”. Also, line 357, “…damaged samples were measured again and they were then normalized to the values of the initial samples without damage.”. Also, title of Fig.11, “…nonlinearity parameters A_/AaAb versus longitudinal engineering strain, superimposed on...”.

10.   Conclusions. Please consider rewriting the conclusions.

Author Response

Response Letter

(Manuscript ID: materials-2221448)

    Thanks very much for the valuable comments on the manuscript materials-2221448 with the title of " Evaluation of plasticity in a thin plate considering the phase-mismatching phenomenon of nonlinear Lamb wave mixing". According to referee’s comments, we made some revisions and supplements. The responses to the referees are listed as followings.

(1) Comments: “…However, the authors state that it was difficult to locate the micro-defects in the thin plate, particularly the important major micro-defect which can grow and cause material failure. Moreover, the size and characteristics of the defect were also not investigated…”

Reply: Thanks for your comments. As mentioned by the reviewer, one unique advantage of the nonlinear Lamb wave mixing is that they can determine the position of micro-damage or defect. In fact, corresponding work can be found in authors’ published papers (Sun, M.; et al. Appl. Phys. Lett. 2019, 114, 011902. and Sun, M. et al. Ultrasonics 2020, 108, 106180.), in which the size and degree of the defect are characterized via the nonlinear Lamb wave mixing. However, we found that the phase-mismatching phenomena usually occur due to the lack of precise acoustic properties for the measured samples, and they may affect the energy transmission as well as reduce the sensitivity to micro-damage. Therefore, the aim of this manuscript is to investigate the influence of the phase-mismatching phenomena on the damage detection.

(2) Comments: “…This present work is based mainly in the article by Pruell et al. (2009): A nonlinear-guided wave technique for evaluating plasticity-driven material damage in a metal plate. NDT&E International 42 (2009) 199–203…Furthermore, the literature review in the introduction is missing the work by Lemaitre (1985), modeling the development of material damage due to plastic deformations: Lemaitre, J., 1985, A continuous damage mechanics model for ductile fracture. Journal of Engineering Materials and Technology. vol. 107: 83-89…”

Reply: Thanks, it is a good comment. Two papers are referenced in the revised manuscript. However, the second harmonics are utilized for assessing the plasticity in the mentioned paper (Pruell. C.; et al. NDT&E Int. 2009, 42, 199-203.), which is different with the methods used in our manuscript.

(3) Comments: “…Therefore, the presented method to evaluate the development of material micro-damage due to plastic deformations by ultrasound waves are somewhat limited. For future work, I would suggest to correlate the signal amplitude ratio A±/(Aa.Ab), which satisfy the elastic resonance conditions, with the experimental variation of Young’s elastic modulus due to plastic deformations…”

Reply: Thanks for the suggestions. In the future work, we will pay more attention on the correlation between A±/(Aa∗Ab) and variation of Young’s modulus in experimental measurements.

(4) Comments: “Title, I suggest to correct “Evaluation of plasticity damage in a thin plate, using the phase-mismatching phenomenon of nonlinear Lamb elastic wave mixing”.”

Reply: Thanks, it is a good comment. In fact, the influence of phase-mismatching phenomena on damage detection is systematically investigated in our manuscript, but only the mode triplet that exactly satisfies the resonance condition is utilized to assess the plasticity. So, I think maybe changing the word ‘‘considering’’ to “using” is not suitable in the title. Finally, based on the reviewer’s comments, the title is revised as “Evaluation of plastic deformation considering the phase-mismatching phenomenon of nonlinear Lamb wave mixing”.

(5) Comments: “Abstract, line 15, please correct “Nonlinear guided elastic waves…”. Also, line 29, please correct “…assessing the accumulated plastic deformations in the thin plates...”.”

(6) Comments: “Introduction, pg.1, please correct “Loads that exceed the yield stress limit cause irreversible plastic deformations…”. Also, line 37, “…may occur during the plastic deformation.”. Also, line 38, “…These undesirable inhomogeneous and localized damage...”.”

(7) Comments: “Introduction, pg.2, line 47, please correct “Recently, several novel methods were proposed…”. Also, correct “were” throughout the text. Also, line 48, “…So mode (lowest-order symmetric Lamb elastic waves)…”. Also, correct “Lamb elastic waves” throughout the text.”

(8) Comments: “Introduction, pg.2, line 53, please correct “…between the material inhomogeneity and plastic deformations…”. Please correct “material inhomogeneity” throughout the text. Also, line 54, please correct “However, the nonlinearities or signal noise interference in measurement systems…”. Also, “…as discussed by Pruell et al. [8,9] and…” please include the reference: [9] Pruell et al. (2009): A nonlinear-guided wave technique for evaluating plasticity-driven material damage in a metal plate. NDT&E International 42 (2009) 199–203.”

(9) Comments: “Theoretical background, pg.3, line 115, please correct “…denotes the through-thickness displacements…”. Also, pg.5, line 155, “…may lead to the variations of the Young’s modulus [1] and TOECs.”. Also, pg.5, line 166, “…excited due to their almost…”. [1] Lemaitre, J., 1985, A continuous damage mechanics model for ductile fracture. Journal of Engineering Materials and Technology. vol. 107: 83-89.”

(10) Comments: “Theoretical background, pg.5, line 172, please correct “…longitudinal elastic waves [37]”.”

Reply: Thanks for these comments. As suggested, we have revised all the mentioned sentences and added two papers in our manuscript.

(11) Comments: “Numerical investigation of nonlinear Lamb waves mixing, pg.5, line 192, “Corresponding third-order elastic constants (TOECs) are also involved here, in which A, B, and C are equal to…” please define these material elastic constants ? ”

Reply: Thanks for your suggestions. The third-order elastic constants (TOCEs) A, B, and C are defined by the Landau and Lifshitz [43], whose relations with TOCEs used by Murnaghan [44] can be written as l=B+C, m=A/2+B, and n=C. The book titled “Theory of elasticity” and paper titled “Finite deformations of an elastic solid” are utilized as the reference [43] and 44] in the revised manuscript.

More details can see Line 195-197 in Page 5.

(12) Comments: “Characterization of the plastic deformation in a thin plate, pg.11, line 346, please correct “Six dog-bone samples were cut from…”. Also, line 350, “…of these undamaged samples were obtained…”. Also, line 353, “…samples were subjected to...”. Also, “...above the yield stress limit via…”. Also, line 355, “…maximum engineering strains of…”. Also, line 357, “…damaged samples were measured again and they were then normalized to the values of the initial samples without damage.”. Also, title of Fig.11, “…nonlinearity parameters A_/AaAb versus longitudinal engineering strain, superimposed..”.”

Reply: Thanks. As suggested, all the mentioned sentences are revised in our manuscript.

(13) Comments: “Conclusions. Please consider rewriting the conclusions.”

Reply: Thanks. As suggested, A novel structure of the conclusion is built and the contents of the conclusion is simplified in the revised manuscript.

Thanks once again.

Reviewer 3 Report

This is an interesting work; however, before proceeding to the next step, the authors should address the following comments.

1. The language of the manuscript has to be improved.

2. Provide a more in-depth discussion of related previous works.

3. In the “Conclusion” section, I recommend presenting more quantitative data as the main results of the research study.

Author Response

Response Letter

(Manuscript ID: materials-2221448)

    Thanks very much for the valuable comments on the manuscript materials-2221448 with the title of "Evaluation of plasticity in a thin plate considering the phase-mismatching phenomenon of nonlinear Lamb wave mixing ". According to referee’s comments, we made some revisions and supplements. The responses to the referees are listed as follows.

(1) Comments: “The language of the manuscript has to be improved.”

Reply: Thanks for your comments. The manuscript is checked by a native speaker and more revised details can be seen in the submitted files.

(2) Comments: “Provide a more in-depth discussion of related previous works.”

Reply: Thanks, it is a good comment. As suggested, the application of the nonlinear Lamb wave mixing on composite laminates is involved. Furthermore, the organization of this paper is described at the end of Section 1.

(3) Comments: “In the “Conclusion” section, I recommend presenting more quantitative data as the main results of the research study.”

Reply: Thanks for the suggestions. As suggested, A novel structure of the conclusion is built and more details are added in the revised manuscript.

More details can see Line 454-478 in Page 14 and 15.

Thanks once again.

Round 2

Reviewer 2 Report

The article is accepted with the corrections.

For future work, I would suggest to correlate the wave signal amplitude ratio A±/(Aa.Ab), which satisfy the elastic resonance conditions, with the experimental variation of Young’s elastic modulus due to plastic deformations.